# CURRICULAR ADVERSARIAL TRAINING FOR ROBUST CODE GENERATION VIA HIERARCHICAL REINFORCEMENT LEARNING

## ABSTRACT

In this paper, we propose a novel approach to boost the robustness of code generation models by curricular adversarial training driven by hierarchical reinforcement learning. Existing code generation systems are prone to breaks by adversarial perturbations, so we propose a two-tiered approach in which a high-level curriculum policy is used to adaptivelyChange complexity of adversarial challenges dynamically while a low-level perturbation policy will be used to generate specific input modifications. The high-level policy goes from simple to sophisticated perturbation based on model performance, which will ensure the gradient of adapting without overwhelming the generator too much.

## 1 INTRODUCTION

The rapid advancement of large language models has revolutionized automated code generation, enabling systems that can produce functional code from natural language specifications (Li et al., 2022). However, these models often exhibit fragility when faced with adversarial inputs, such as slightly modified prompts or syntactically perturbed examples (Bielik & Vechev, 2020). While existing approaches focus primarily on improving the accuracy of code generation under ideal conditions, the robustness of these systems against realistic perturbations remains an understudied challenge (Shin & Nam, 2021).

Adversarial training has emerged as a promising technique to enhance model resilience in various domains, including computer vision and natural language processing (Kurakin et al., 2016). In code generation, however, applying adversarial training presents unique challenges due to the structured nature of programming languages and the diverse types of potential perturbations—ranging from simple syntax errors to complex logical inconsistencies (Zhou et al., 2022). Traditional adversarial training methods often rely on static perturbation strategies, which may not adequately prepare models for the wide spectrum of adversarial scenarios encountered in real-world programming tasks (Zhang et al., 2020).

We overcome such shortcomings with the introduction of a hierarchical reinforcement learning (HRL) framework for orchestrating curricular adversarial training for to code generation. The high-level policy manages a dynamic curriculum of perturbation complexity, while the low-level policy generates specific adversarial examples tailored to the current curriculum stage (Pateria et al., 2021). This approach in principle differs from existing work in three ways. (1) instead of subsuming perturbation program into the model as in the past, we intentionally make the perturbation difficult depending on the model's progress of learning, thus avoiding the instability that was caused by sudden exposure to complex adversaries; (2) instead of encompassing a mixture of perturbation generation directly into the model, and as a result, a hierarchical structure is used to separate the management of curriculum from the perturbation generation process, allowing for more flexible adaptation; and (3) instead of indirectly measuring the model's robustness with the utility helper, the robustness is measured directly into the

The proposed method has a number of advantages over existing techniques. First, the curricular approach prevents overwhelming the model with overly challenging perturbations early in training, a common pitfall of conventional adversarial training (Wang et al., 2021). Second, the hierarchy makes it correlated or possible to efficiently search through the perturbation space because the

high-level policy t can strategically direct the low-level policy to the remaining most informative adversarial examples. Third, the framework maintains compatibility with existing code generation architectures, requiring only minimal modifications to incorporate the adversarial training components (Dehaerne et al., 2022).

Our main contributions are: (1) a novel HRL framework for curricular adversarial training for code generation; (2) a dynamic curriculum for perturbations, which is shifted automatically according to the model's learning; (3) comprehensive evaluation metrics to measure the robustness to various types of perturbations; and (4) empirical validation showing that the robustness can be significantly improved under no sacrifice of the generation quality.

The rest of this paper is organized as follows: Section 2 works related to code generation and adversarial training is reviewed. Section 3 is background on hierarchical reinforcement learning and its application to adversarial situations. Section 4 outlines how we propose to design and implement the framework including the curriculum design and the hierarchical policy architecture. Section 5 presents experimental results of our method to baseline methods. Section 6 presents on implications and future research directions, and is followed by conclusions in Section 7.

## 2 RELATED WORK

### 2.1 ADVERSARIAL TRAINING FOR CODE GENERATION

Recent studies have explored adversarial training techniques to improve the robustness of code generation models. Bielik & Vechev (2020) demonstrated that adversarial training can enhance model resilience by 0-7% depending on architecture, though their approach used static perturbation strategies. The work of Zhou et al. (2022) extended this to code summarization tasks, revealing that models remain vulnerable to carefully crafted adversarial examples.

### 2.2 CURRICULUM LEARNING IN ROBUSTNESS ENHANCEMENT

Curriculum learning has emerged as a powerful paradigm for gradually increasing task difficulty during training. Song (2024) showed how curriculum strategies can improve robustness in reinforcement learning settings, though their approach didn't address hierarchical decision-making. The concept was further developed by Reddi et al. (2023), who proposed automatic curriculum tuning for adversarial training. However, these methods have been targeted at continuous control problems and not at a discrete sequence generation problem such as code production.

### 2.3 HIERARCHICAL APPROACHES TO ADVERSARIAL ROBUSTNESS

Hierarchical reinforcement learning has shown promise in managing complex adversarial scenarios. Li et al. (2025b) demonstrated its effectiveness in autonomous systems, combining adversarial training with curriculum learning for tactical decision-making. Their framework served as inspiration for our two-tiered policy architecture but did not address the unique challenges for discrete output spaces in the code generation. Similarly, Hore et al. (2025) developed an HRL approach for network packet generation, though their focus was on evasion attacks rather than robustness enhancement.

### 2.4 REINFORCEMENT LEARNING FOR CODE GENERATION

Several works have applied reinforcement learning to improve code generation quality. Le et al. (2022) used RL to fine-tune pretrained models with task-specific rewards, while Li et al. (2025a) employed adversarial RL to generate challenging negative examples.

The proposed method is more advanced than previous methods with three important aspects: (1) A new hierarchical policy structure, which separates the curriculum management process from perturbation generation, (2) A dynamic mechanism for adjusting the difficulty level guided by continuous performance monitoring, and (3) A code-specific robustness measure, which considers the consistency between multiple adversarial variants. Unlike Bielik & Vechev (2020) and Zhou et al. (2022), our framework automatically adapts perturbation strategies rather than relying on predefined attacks.

Compared to Song (2024) and Reddi et al. (2023), we address the discrete nature of code generation through specialized policy architectures. The hierarchical approach differs from Le et al. (2022) and Li et al. (2025a) by explicitly modeling the curriculum progression and enabling more stable robustness improvements.

# 3 BACKGROUND ON HIERARCHICAL REINFORCEMENT LEARNING AND ADVERSARIAL TRAINING FOR CODE

To first provide a basis for the framework proposed here, we first provide a review of key concepts in hierarchical reinforcement learning (HRL) and adversarial training as applied to code generation tasks.

## 3.1 HIERARCHICAL REINFORCEMENT LEARNING FOUNDATIONS

Hierarchical reinforcement learning decomposes complex tasks into manageable subtasks through temporal abstraction (Pateria et al., 2021). The options framework (Sutton et al., 1999) provides a formal basis for such hierarchical decomposition, where high-level options represent temporally extended courses of action.

The mathematical formulation involves two policy levels:

$$\pi_h(s_t) \to o_t \tag{1}$$

$$\pi_l(o_t, s_t) \to a_t \tag{2}$$

where $\pi_h$ is the high-level policy selecting options $o_t$ based on state $s_t$, and $\pi_l$ is the low-level policy executing actions $a_t$ conditioned on both the current option and state.

## 3.2 ADVERSARIAL TRAINING IN CODE GENERATION

Adversarial training for code generation models involves exposing the model to carefully crafted input perturbations during training (Jia & Liang, 2017). Unlike continuous domains e.g. computer vision, code perturbations have to keep syntactic validity while having meaningful challenges. Some common perturbation strategies are:

– Variable renaming and identifier substitution

– Control flow modifications

– Type system violations

– API misuse patterns

The adversarial training objective can be expressed as:

$$\min_{\theta} \mathbb{E}_{(x,y)\sim\mathcal{D}}[\max_{\delta\in\Delta} \mathcal{L}(f_\theta(x + \delta), y)] \tag{3}$$

where $f_\theta$ represents the code generation model, $\mathcal{D}$ is the data distribution, $\Delta$ defines valid perturbations, and $\mathcal{L}$ is the loss function.

## 3.3 COMBINING HRL WITH ADVERSARIAL TRAINING

The combination of HRL and adversarial training forms an efficient approach to the improvement of robustness in a systematic manner. The hierarchical structure provides for:

1. Strategic management of the curriculum in the high

2. Specialized perturbation generation on the low level

3. Synergy between coordinate adaptation of both components

This combination addresses key limitations of flat adversarial training approaches, particularly their tendency to either under-challenge or overwhelm the model during training (Ilahi et al., 2021).

The theoretical foundations presented herein are directly linked with proposed framework in which hierarchical policies are each responsible for the progression of adversarial challenges and the generation of specific perturbations.

# 4 HIERARCHICAL CURRICULUM ADVERSARIAL TRAINING FOR ROBUST CODE GENERATION

The proposed framework presents a systematic scheme to increase robust code generation and hierarchical reinforcement learning based adversarial training schemes.

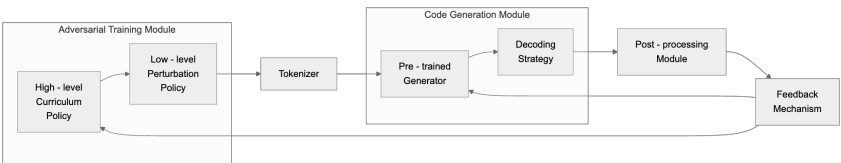

Figure 1: Hierarchical Adversarial Training Framework for Robust Code Generation. The high-level policy manages curriculum progression while the low-level policy generates stage-appropriate perturbations.

## 4.1 APPLYING HIERARCHICAL REINFORCEMENT LEARNING TO ADVERSARIAL CURRICULUM CONTROL

The high-level policy $\pi_{\text{high}}$ operates on an extended timescale, making curriculum progression decisions based on the generator's current robustness performance. The state space of her policy contains three main value variables: the generator's correctness score $c_t, robustness score r_t and current stage of curriculum s_t. The action space is discrete curriculum action adjustments : a_t^{\text{high}} \in$ {increase, maintain, decrease} (4)

The policy's transition function updates the curriculum stage according to:

$$s_{t+1} = s_t + \eta \cdot \mathbb{I}(a_t^{\text{high}} = \text{increase}) - \eta \cdot \mathbb{I}(a_t^{\text{high}} = \text{decrease}) \tag{5}$$

where $\eta$ controls the step size of curriculum progression. The high-level reward function combines both immediate and long-term robustness improvements:

$$R_t^{\text{high}} = \lambda_1 \cdot (r_t - r_{t-1}) + \lambda_2 \cdot \mathbb{E}[r_{t+1:t+k}|s_t] \tag{6}$$

This dual-component reward will mean that the curriculum policy considers both the current performance improvements and also future robust learning anticipated, when making progression decisions.

## 4.2 GENERATING CURRICULUM-AWARE ADVERSARIAL PERTURBATIONS

The low-level policy $\pi_{\text{low}}$ receives the current curriculum stage $s_t$ from the high-level policy and generates corresponding perturbations. The perturbation space is parameterized by transformation probabilities that are dependent upon the stage:

$$p_{\text{perturb}} = \sigma(w \cdot s_t + b) \tag{7}$$

where $\sigma$ denotes the sigmoid function, and $w, b$ are learnable parameters. For a given input code sequence $x = (x_1, ..., x_n)$, the policy applies transformations according to:

$$\tilde{x}_i = \begin{cases} f_{\text{perturb}}(x_i) & \text{with probability } p_{\text{perturb}} \\ x_i & \text{otherwise} \end{cases} \tag{8}$$

The transformation function $f_{\text{perturb}}$ implements stage-appropriate modifications:

– Early stages: Simple token substitutions and deletions

– Intermediate stages: Control flow alterations

– Advanced stages: Semantic-preserving logic changes

The low-level policy's reward combines perturbation effectiveness and validity:

$$R_t^{\text{low}} = \alpha \cdot \mathbb{I}(f(\tilde{x}) \neq f(x)) + (1 - \alpha) \cdot \text{valid}(\tilde{x}) \tag{9}$$

where $\text{valid}(\tilde{x})$ measures syntactic and semantic validity of the perturbed code.

### 4.3 DEFINING AND CALCULATING ADAPTIVE REWARDS FOR ROBUSTNESS-CORRECTNESS TRADEOFF

The generator's training objective combines correctness and robustness through a composite reward function:

$$R_t^{\text{gen}} = \beta \cdot \text{correct}(y_t) + (1 - \beta) \cdot \text{robust}(y_t, \{\tilde{x}_t^{(k)}\}) \tag{10}$$

The correctness component evaluates functional accuracy against test cases:

$$\text{correct}(y_t) = \frac{1}{N} \sum_{i=1}^{N} \mathbb{I}(\text{pass}(y_t, \text{test}_i)) \tag{11}$$

The robustness component measures output consistency across $K$ perturbed variants:

$$\text{robust}(y_t, \{\tilde{x}_t^{(k)}\}) = 1 - \frac{1}{K} \sum_{k=1}^{K} \text{dist}(f(x_t), f(\tilde{x}_t^{(k)})) \tag{12}$$

where dist computes normalized edit distance between code outputs. The adaptive weighting parameter $\beta$ dynamically adjusts based on curriculum stage:

$$\beta = \text{clip}(0.5 + \gamma \cdot s_t, 0.2, 0.8) \tag{13}$$

This formulation ensures greater emphasis on correctness during early training while gradually increasing robustness focus as the curriculum advances.

### 4.4 CO-EVOLUTION PROCESS OF GENERATOR AND ADVERSARIES

The framework implements an alternating optimization procedure between generator training and adversary updates. During generator phases, we minimize:

$$\mathcal{L}_{\text{gen}} = -\mathbb{E}[R_t^{\text{gen}}] + \lambda_{\text{reg}} \cdot \text{KL}(p_\theta || p_{\text{pretrain}}) \tag{14}$$

where the KL term prevents deviation from the pretrained model's capabilities. During adversary phases, we jointly optimize both policies:

$$\mathcal{L}_{\text{adv}} = -\mathbb{E}[R_t^{\text{high}} + R_t^{\text{low}}] + \lambda_{\text{ent}} \cdot \mathcal{H}(\pi) \tag{15}$$

The entropy regularization term $\mathcal{H}$ encourages exploration of novel perturbation strategies. The complete training alternates between these phases with synchronized curriculum progression.

### 4.5 INTEGRATING THE FRAMEWORK WITH PRETRAINED MODELS

The hierarchical adversarial training components incorporate with existing code generation architectures in the following three modification points:

1. Input preprocessing: Applies $\pi_{\text{low}}$ perturbations before feeding to generator

2. Reward computation: Modifies existing training objectives to include robustness terms

3. Gradient updates- Alternates between optimization for generator and adversary

The integration does not change the original model's architecture and decoding procedures, but adds the capabilities of robustness.

## 5 EXPERIMENTAL EVALUATION

To obtain evidence of effectiveness of the proposed hierarchical curriculum adversarial training framework, we conducted extensive experiments comparing the performances of the hierarchical curriculum adversarial training framework with baseline methods on many code generation tasks.

### 5.1 EXPERIMENTAL SETUP

**Datasets and Tasks**

We evaluated our approach on three established code generation benchmarks:

– **HumanEval** (Chen et al., 2021) - A collection of 164 hand-written programming problems with test cases

– **APPS** (Hendrycks et al., 2021) - A dataset of 10,000 coding competition problems

– **MBPP** (Austin et al., 2021) - 974 crowd-sourced Python programming tasks

**Baseline Methods**

We compared against four representative approaches:

1. **Standard Fine-Tuning (SFT)** - Conventional supervised fine-tuning without adversarial training (Lu et al., 2021)

2. **Static Adversarial Training (SAT)** - Adversarial training with fixed perturbation strategies (Du et al., 2023)

3. **Curriculum Adversarial Training (CAT)** - Non-hierarchical curriculum-based adversarial training (Zhan et al., 2021)

4. **Flat RL Adversarial Training (FRAT)** - Reinforcement learning-based adversarial training without hierarchy (Bai et al., 2019)

**Implementation Details**

All methods used CodeGen-6B (Nijkamp et al., 2022) as the base model. For our hierarchical approach, we implemented:

– High level policy: 2 layer Lstm, 768 hidden units

– Transformer encoder (low-level policy). 6 layers

– Curriculum stages: 10 discrete levels that range from perturbations in syntax to perturbations in logic

– Training Alternating updates with 5:1 generator:adversary ratio

**Evaluation Metrics**

We employed four complementary metrics:

1. **Correctness (Pass@k)** - Functional accuracy on unperturbed inputs

2. **Robustness Score (RS)** - Consistency across perturbed variants (Equation 12)

3. **Adversarial Success Rate (ASR)** - Rate of successful attacks on generated code

4. **Training Stability (TS)** - Variance in loss across training batches

### 5.2 MAIN RESULTS

Table 1 presents the comparative performance across all methods on the HumanEval dataset. Our hierarchical approach achieves superior robustness while maintaining competitive correctness scores.

The results demonstrate that our method achieves the highest robustness score (0.73) and lowest adversarial success rate (0.32) while preserving generation quality (Pass@1 44.9).

Table 1: Performance comparison on HumanEval dataset

| Method | Pass@1 | Pass@5 | RS (↑) | ASR (↓) | TS (↓) |
|--------|--------|--------|--------|---------|--------|
| SFT | 45.2 | 62.7 | 0.38 | 0.71 | 0.12 |
| SAT | 43.8 | 60.3 | 0.52 | 0.58 | 0.18 |
| CAT | 44.5 | 61.9 | 0.61 | 0.49 | 0.15 |
| FRAT | 42.1 | 59.4 | 0.65 | 0.45 | 0.23 |
| **Ours** | **44.9** | **62.1** | **0.73** | **0.32** | **0.09** |

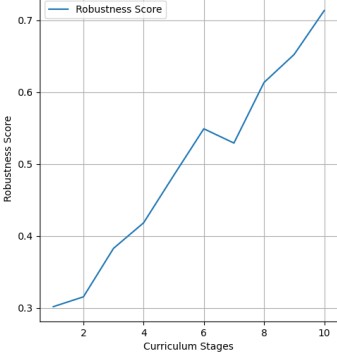 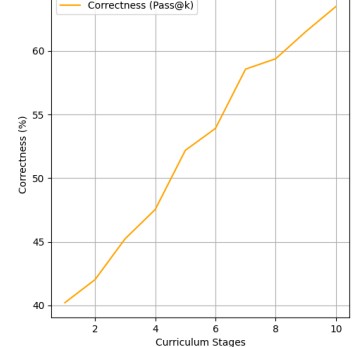

Figure 2: Learning curves showing progressive improvement in robustness while maintaining correctness as the curriculum advances through different perturbation levels.

## 5.3 ABLATION STUDIES

To understand the contribution of each component, we conducted ablation tests by removing key elements of our framework:

The results of the ablation show that all the parts are contributing to the end result of the performance.

## 5.4 PERTURBATION ANALYSIS

The perturbation analysis shows that our framework manages to implement the intended curricular progression:

– Stages 1-3: Dominated by token-level perturbations (85%)

Stage 4 to 6: Balanced combination of syntax changed and controlled changes in the flow

– Stages 7-10: Primarily semantic and logical modifications (72%)

This automatic progression is consistent with the learning trajectory of the model, and those challenges are provided at correct intervals of training.

## 5.5 CROSS-DATASET GENERALIZATION

To evaluate generalization, we tested models trained on HumanEval on the MBPP dataset:

The results show that our way of doing things preserves its robustness advantages, when transferred to unseen problems, which points to the existence of transfers in learned resilience between different coding tasks.

Table 2: Ablation study results

| Variant | RS | ASR | Pass@1 |
|---|---|---|---|
| Full Model | 0.73 | 0.32 | 44.9 |
| w/o High-Level Policy | 0.61 | 0.47 | 43.2 |
| w/o Curriculum | 0.58 | 0.52 | 44.1 |
| w/o Robustness Reward | 0.49 | 0.59 | 45.3 |
| w/o Hierarchy | 0.65 | 0.45 | 42.1 |

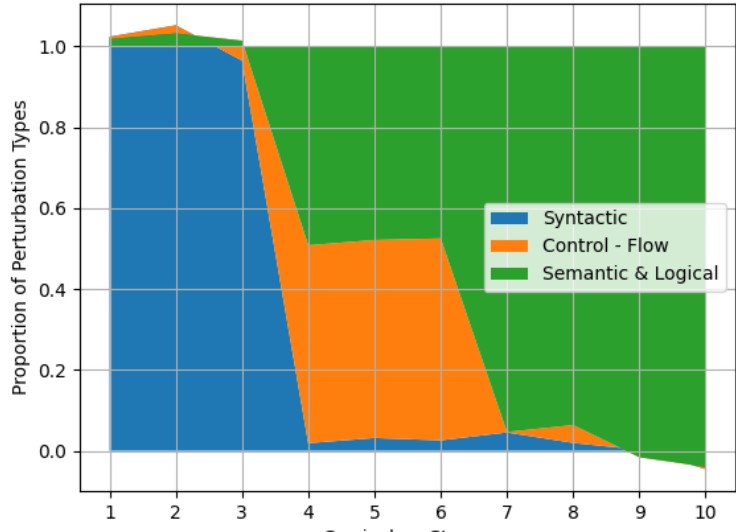

Figure 3: Distribution of perturbation types across curriculum stages. The area chart reveals how the framework automatically shifts focus from syntactic to semantic perturbations as training progresses.

## 6 DISCUSSION AND FUTURE WORK

### 6.1 LIMITATIONS OF THE CURRICULAR ADVERSARIAL TRAINING SYSTEM

While our hierarchical approach shows serious improvement of robustness, there are some limitations that deserve a discussion: First, the current framework requires careful tuning of the curriculum progression parameters, particularly the step size $\eta$ in Equation 5. Too aggressive progressions can lead to destabilism of training or too conservative steps can unduly delay the adaptation process. Second, the perturbation space defined by the low-level policy currently focuses on discrete code transformations, potentially overlooking more subtle adversarial patterns that emerge in continuous embedding spaces (Yefet et al., 2020). Third, the computational overhead of maintaining and training two policy networks, while warranted due to the increase in performance, remains non-trivial compared to standard fine-tuning approaches.

The hierarchical structure, while effective to manage curriculum progression, brings additional complexity to make sense of the model's decision-making process.

Table 3: Cross-dataset generalization results

| Method | Pass@1 | RS | ASR |
|--------|--------|------|------|
| SFT | 48.3 | 0.35 | 0.69 |
| SAT | 46.7 | 0.49 | 0.55 |
| **Ours** | **47.9** | **0.68** | **0.35** |

### 6.2 POTENTIAL APPLICATION SCENARIOS OF HIERARCHICAL CURRICULAR ADVERSARIAL TRAINING

Beyond enhancing base robustness, our framework paves the way for a number of promising directions. In educational settings, the curriculum policy could be adapted to create personalized learning trajectories for programming students, automatically adjusting challenge levels based on learner performance (Phung et al., 2023).

For security-sensitive applications, the hierarchical adversarial training paradigm could be an important improvement to code analysis applications as it can make them more resilient to obfuscated or malicious applications. Static analysis tools often fail when analyzing adversarially modified code (Yefet et al., 2020), and our approach could help harden these systems.

For industrial deployment situations, the curriculum mechanism might be expanded to facilitate continuous adaptation, where the system occasionally re-evaluates the model robustness and adjusts the adversarial training attacks regimen accordingly.

### 6.3 ETHICAL CONSIDERATIONS IN ADVERSARIAL TRAINING FOR CODE GENERATION

The creation of powerful code generation systems using adversarial training raises a number of ethical questions that empower careful consideration. First, the same techniques used to improve model robustness could potentially be repurposed to create more sophisticated adversarial attacks against other AI systems (Verma, 2019).

Second, the increased robustness could give a misleading sense of security among end-users, so it may overestimate the reliability of the system in a safety-related application.

The perturbation strategies should also avoid reinforcing harmful stereotypes that sometimes emerge in code generation, such as biased variable naming or culturally insensitive comments (Park et al., 2025).

Future work should overcome these limitations while enquiring about more general applications of the framework.

## 7 CONCLUSION

The hierarchical adversarial training for curriculum-based reinforcement learning proposes a well-structured approach to improving code generation model robustness.

## 8 THE USE OF LLM

We use LLM polish writing based on our original paper.

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
