# OpenReview forum: "Curricular Adversarial Training for Robust Code Generation via Hierarchical Reinforcement Learning"
_ICLR.cc/2026/Conference — Submitted to ICLR 2026_

### Official Review · Reviewer_TNBK · 2025-10-31

**Soundness:** 1
**Presentation:** 1
**Contribution:** 1
**Rating:** 0
**Confidence:** 5

**Summary:**

The paper proposes a hierarchical reinforcement learning (HRL) framework for curricular adversarial training in code generation. The method introduces two policies, i.e., a high-level policy that dynamically adjusts the complexity of adversarial perturbations, and a low-level policy that generates specific adversarial examples. The goal is to improve the robustness of large language models for code generation against adversarial perturbations. Experiments are conducted on HumanEval, APPS, and MBPP datasets using CodeGen-6B as the base model.

**Strengths:**

1. The general idea of combining curriculum learning with HRL for adversarial robustness is conceptually sound and has potential.

2. The paper attempts to unify curriculum scheduling, hierarchical control, and adversarial training into a single framework.

**Weaknesses:**

1. The manuscript suffers from many typos, grammatical mistakes, and formatting inconsistencies. The writing style lacks clarity and precision. Several sentences appear incomplete or poorly constructed (e.g., Section 1 and 4). The overall presentation resembles a coursework report rather than a polished conference paper.

2. The paper does not clearly define the adversarial setting for code generation—what kinds of perturbations are used, how they are measured, and what “robustness” precisely means. The motivation for using hierarchical RL in this context is not well justified compared to simpler baselines.

3. The experimental section is extremely limited. Only high-level summaries are given—no implementation details, hyperparameters, or reproducibility discussion. The results appear arbitrary and lack statistical analysis. There are no significance tests or comparisons beyond simple table metrics. Ablation studies and cross-dataset evaluations are superficial and not convincing.

**Questions:**

This paper is not ready for publication. It reads more like a course project or exploratory report rather than a rigorously developed research paper.

---

### Official Review · Reviewer_8dGk · 2025-11-01

**Soundness:** 3
**Presentation:** 2
**Contribution:** 2
**Rating:** 2
**Confidence:** 3

**Summary:**

This paper uses curricular adversarial training and hierarchical reinforcement learning for code generation tasks. While the idea is sound, related works is good, and the overall structure for experiments seems solid, the paper seems a bit underdeveloped.

**Strengths:**

The underlying structure of the paper is sound. Code generation is an active area of research that is not yet fully solved. Adversarial curricular learning seems solid. Hierarchical RL also makes sense since most coding problem themselves are a form of hierarchical problem (class, function, etc). The algorithm was compared with some baselines as well.

**Weaknesses:**

The paper up to introduction and related works seems OK, but the subsequent sections seems less developed. For example, it appears there's a typo in the Line 189 that looks like an LaTEX formatting issue. While bullet-pointing makes idea very clear, I think it would be better if there were more descriptions of the idea. For example, how exactly does the reward computation works here in section 4.5? How is robustness term calculated and how is it incorporated as? How are the gradient updates calculated and implemented? The figures are in curriculum stages, but how is it in terms of computational times or steps? There isn't enough details to more accurately assess the algorithm.

There are other stylistic issues as well. For example there's an typo at the abstract (adaptivelyChange). Text in figure 2 is too small to be legible. While Figure 2 surves as a good hierarchy, it might be nice to incorporate a pseudocode algorithm as well for a bit more clarity with nomenclature as well. Also, most LLM papers uses SFT as 'Supervised Fine-Tuning' so using SFT as 'Standard Fine-Tuning' may mislead some readers.Finally, In Figure 3, the porportion of perturbed types reaches higher than 1 at stage 1 throguh 3, below zero at stages 9 and 10, which seems like an visualization error.

**Questions:**

This paper seems incomplete and should be rewritten.

---

### Official Review · Reviewer_zv9A · 2025-11-02

**Soundness:** 1
**Presentation:** 1
**Contribution:** 1
**Rating:** 0
**Confidence:** 4

**Summary:**

The writing in this paper is poor and often difficult to follow, with numerous grammatical issues, incomplete sentences, missing definitions, and unclear explanations throughout. As a result, it is challenging to fully understand the proposed method, and the following summary reflects my interpretation to the best of my understanding.

This paper proposes a hierarchical reinforcement learning (HRL) framework for curricular adversarial training in code generation. A high-level policy controls the progression of adversarial perturbation complexity, while a low-level policy generates the specific perturbations.

**Strengths:**

- studies an important topic: improving the robustness of code generation models under adversarial perturbations.

**Weaknesses:**

The paper is difficult to follow, with numerous grammatical errors, awkward phrasing, and undefined notations (e.g., Line 16, 50, 190, to name just a few). Several sections (e.g., Section 4) are nearly unreadable, containing undefined equations and terms, typos, and long, unstructured phrases that make the overall presentation very hard to read.

**Questions:**

There are many things unclear in the current presentation. I think the paper will benefit from a complete rewriting to explain all aspects clearly. For example, how are the high-level and low-level policies trained jointly? How are adversarial perturbations for code defined and validated to maintain syntactic and semantic correctness? Can you provide concrete examples of adversarial perturbations at different curriculum stages? What is the rationale for using hierarchical RL, and how does it improve over simpler curriculum adversarial training baselines?

---

### Meta-Review · Area_Chair_WWxj · 2025-12-16

**Summary:**

The manuscript reads like an early draft: typos, inconsistent formatting, and half-finished sentences (especially in Sections 1 and 4) accumulate faster than the reader can forgive. The prose feels closer to a course assignment than to conference-ready work.

More importantly, the adversarial setup itself is never rigorously specified. The authors never say what a “perturbation” looks like for code, how its magnitude is calibrated, or what formal guarantee (or even informal notion) of robustness is being pursued. Likewise, the leap to hierarchical RL is taken on faith; no argument is offered for why two-layer control is preferable to flatter, simpler strategies that curate the same curriculum.

Experiments are equally skeletal. The reader is shown only glossy summary numbers—no hyper-parameters, no random-seed protocol, no link to runnable artifacts, and no statistical scrutiny. The handful of tables omit error bars, significance tests, or any serious baseline beyond the authors’ own ablations, leaving the empirical contribution impossible to verify or extend.

**Reviewer Concerns:**

The authors did not respond, and the reviewers' comments remain completely unaddressed.

**Reviewer Scores:**

The reviewers consistently maintained negative scores.

---

### Decision · Program_Chairs · 2026-01-26

Reject